# *Myristica fragrans* Extract Regulates Gut Microbes and Metabolites to Attenuate Hepatic Inflammation and Lipid Metabolism Disorders via the AhR–FAS and NF-κB Signaling Pathways in Mice with Non-Alcoholic Fatty Liver Disease

**DOI:** 10.3390/nu14091699

**Published:** 2022-04-19

**Authors:** Wenyu Zhao, Min Guo, Jun Feng, Zhennan Gu, Jianxin Zhao, Hao Zhang, Gang Wang, Wei Chen

**Affiliations:** 1State Key Laboratory of Food Science and Technology, Jiangnan University, Wuxi 214122, China; zwyjiangnan@163.com (W.Z.); guomin@jiangnan.edu.cn (M.G.); zhennangu@jiangnan.edu.cn (Z.G.); zhaojianxin@jiangnan.edu.cn (J.Z.); zhanghao@jiangnan.edu.cn (H.Z.); weichen@jiangnan.edu.cn (W.C.); 2School of Food Science and Technology, Jiangnan University, Wuxi 214122, China; 3Department of Ultrasound, Affiliated Wuxi No. 2 People’s Hospital of Nanjing Medical University, Wuxi 214122, China; 4Yangzhou Institute of Food Biotechnology, Jiangnan University, Yangzhou 225004, China; 5National Engineering Research Center for Functional Food, Jiangnan University, Wuxi 214122, China; 6Wuxi Translational Medicine Research Center and Jiangsu Translational Medicine Research Institute Wuxi Branch, Wuxi 214122, China

**Keywords:** *Myristica fragrans* extract, non-alcoholic fatty liver disease, gut microbes, metabolites, aryl hydrocarbon receptor, nuclear factor kappa B

## Abstract

Recent studies have shown that non-alcoholic fatty liver disease (NAFLD) is closely related to the gut microbiome. *Myristica fragrans* is widely used as a traditional seasoning and has a therapeutic effect on gastrointestinal diseases. Although previous studies have shown that *M. fragrans* extracts have anti-obesity and anti-diabetes effects in mice fed a high-fat diet, few studies have determined the active components or the corresponding mechanism in vivo. In this study, for the first time, an *M. fragrans* extract (MFE) was shown to be a prebiotic that regulates gut microbes and metabolites in mice fed a high-fat diet. Bioinformatics, network pharmacology, microbiome, and metabolomics analyses were used to analyze the nutrient–target pathway interactions in mice with NAFLD. The National Center for Biotechnology Information Gene Expression Omnibus database was used to analyze NAFLD-related clinical data sets to predict potential targets. The drug database and disease database were then integrated to perform microbiome and metabolomics analyses to predict the target pathways. The concentrations of inflammatory factors in the serum and liver, such as interleukin-6 and tumor necrosis factor-α, were downregulated by MFE. We also found that the hepatic concentrations of low-density lipoprotein cholesterol, total cholesterol, and triglycerides were decreased after MFE treatment. Inhibition of the nuclear factor kappa B (NF-κB) pathway and downregulation of the fatty acid synthase (FAS)-sterol regulatory element-binding protein 1c pathway resulted in the regulation of inflammation and lipid metabolism by activating tryptophan metabolite–mediated aryl hydrocarbon receptors (AhR). In summary, MFE effectively attenuated inflammation and lipid metabolism disorders in mice with NAFLD through the NF-κB and AhR–FAS pathways.

## 1. Introduction

Non-alcoholic fatty liver disease (NAFLD) is a clinicopathological syndrome characterized by the abnormal accumulation of triglycerides in hepatocytes in the absence of excessive alcohol consumption. Currently, there are no specific drugs that effectively prevent or treat NAFLD. Therefore, studying the pathogenesis of NAFLD and exploring possible therapeutic targets will contribute to its prevention and treatment [1,2]. The occurrence of NAFLD may be related to many factors. In addition to the classic ‘double-hit’ theory (steatosis is the first hit, and inflammation is the second hit necessary to develop NASH.) and the ‘multiple-hit’ hypothesis, an imbalance in the intestinal flora is found to be closely related to NAFLD and promotes its progression by regulating metabolites, glycolipid metabolism, immune responses, and inflammation [3].

The gut and the liver are closely linked in structure and function and this relationship is known as the gut–liver axis. The gut–liver axis has been shown to play an important role in the pathogenesis of NAFLD [4]. Increased intestinal permeability in mice fed a high-fat diet is associated with intestinal dysbiosis, which results in an inflammatory reaction and is positively correlated with non-alcoholic steatohepatitis [5]. Some gut microbes and metabolites play a key role in the liver–gut axis. Lactate in the gut has been reported to promote the growth of Bacteroides by enhancing the utilization of polysaccharides [6]. Tryptophan metabolites from gut microbiota suppress the expression of FAS and SREBP-1c in the liver to regulate lipid metabolism by activating hepatic AHR receptors [7].

As the ‘second human genome’ that controls human health, the gut microbiota ferments indigestible food components; synthesizes vitamins and other essential micronutrients; metabolizes dietary toxins, carcinogens, cholesterol, and bile acids; ensures the maturation of the immune system; and prevents colonization by intestinal pathogens [8,9]. Previous studies have shown that the consumption of a high-calorie diet of obesity-causing foods may alter gut microbiota and gut barrier function, thus eliciting an inflammatory response that promotes obesity and NAFLD [10]. Probiotics have different effects on the intestinal flora of NAFLD mice, but all can improve the metabolism of short-chain fatty acids, bile acids, and tryptophan; at the same time, indole compounds produced by the metabolism of intestinal flora have anti-inflammatory in vitro effects and restore the liver weight/body weight ratio, and liver enzyme, cholesterol, and cytokine levels in NAFLD mice [11].A decreased abundance of Akkermansia species in the intestinal mucosa leads to the thinning of the intestinal mucus layer and to an increase in intestinal permeability, leading to leakage of bacteria and their metabolites. Conversely, an increase in the abundance of Akkermansia species improves obesity and NAFLD [12]. The use of probiotics/synbiotics was associated with improvement in liver-specific markers of hepatic inflammation, liver stiffness measurements, and steatosis in persons with NAFLD [13].

Currently, studies have shown that probiotics and prebiotics play important roles in improving intestinal flora homeostasis and thus they have become potential therapies for NAFLD [14,15,16]. Recent pharmacological studies have shown that some herbal monomers or prescriptions may achieve the targeted treatment of NAFLD by regulating the intestinal flora. Berberine is used to treat metabolic diseases, including obesity, NAFLD, and type 2 diabetes, by modulating the gut microbiota [17]. Ganoderma lucidum polysaccharides increase the abundance of probiotics and reduce obesity in mice by modulating the composition of gut microbiota [18]. These findings indicate that herbs, as dietary supplements, play a prebiotic role in the gut.

The seeds of *Myristica fragrans* Houtt (nutmeg), which are used as a traditional seasoning, have a high fat content (28–33%), with myristic acid predominating. The main amino acid in *M. fragrans* Houtt is phenylalanine, which is an essential amino acid [19,20,21]. Numerous active ingredients in *M. fragrans* Houtt have been reported to have various effects, such as anti-inflammatory, antioxidant, and antimicrobial effects [22,23,24,25]. *M. fragrans* Houtt extracts (MFEs) have also been studied in obesity [26], but few studies have addressed their effects on NAFLD via gut microbiota homeostasis and related metabolites. In this study, through multiomics analyses of pharmacology, microbiology, metabolomics, and bioinformatics networks, the main objective was to estimate the function of MFE as a prebiotic, which regulates gut microbes and their metabolites in mice fed a high-fat diet.

## 2. Materials and Methods

### 2.1. Preparation of MFE and Identification of the Main Components

Nutmeg (*M. fragrans* Houtt) seeds were purchased from Beijing Tongrentang (Beijing, China). The seeds were ground into a powder, extracted with 75% ethanol three times, concentrated and dried, and then extracted with petroleum ether three times. The resulting MFE was analyzed using a GGT 0620 integrated two-dimensional gas chromatography-time-of-flight mass spectrometer (Guangzhou Hexin Instrument Co., Ltd., Guangzhou, China). Component identification was performed using the National Institute of Standards and Technology reference database.

### 2.2. Gene Expression Omnibus and Herb Database Mining

NAFLD-related datasets (GSE151158 and GSE57425) were downloaded from the Gene Expression Omnibus (GEO) database (https://www.ncbi.nlm.nih.gov/; 28 December 2021). Differential analysis and screening of the datasets were performed using the GEO2R platform. The GSE151158 dataset included samples from 7 healthy individuals and 17 individuals with moderate fatty liver disease, whereas the GSE57425 dataset included samples from three individuals on a low-fat diet and three individuals on a high-fat diet. Standard criteria (|logFC| > 1; *p* < 0.05) were used to screen for differentially expressed genes (DEGs), with logFC > 1 indicating an upregulated gene, and logFC < 1 indicating a downregulated gene. The two datasets were mapped to identify shared DEGs. Downloads GSE57425 and GSE151158 from the GEO database through the GEO query package removed the probes corresponding to multiple molecules by one probe, and when encountering probes corresponding to the same molecule, only the probe with the largest signal value was kept. The standardization method was as follows: (1) normalization between arrays was performed; (2) the missing value processing used the imputation method to deal with missing data; (3) the clustering situation between sample groups through the PCA chart was checked; and (4) the limma package was used to analyze the differences between the two groups. The specific difference analysis is shown as a heat map.

Drug data were screened using the following criteria: oral bioavailability (OB) ≥ 30%, blood–brain barrier (BBB) permeability ≥ −0.3, drug-likeness (DL) ≥ 0.18, and the target of each MFE component was matched. The UniProt database (https://www.uniprot.org; 30 October 2021) was used to perform further ID conversion and the identification of targets was extracted from the database. After merging the GEO-mined dataset and the data from the herbal database (Traditional Chinese Medicine System Pharmacology Database and Analysis PlatformPharmacology Database and Analysis Platform (TCMSP) and the Encyclopaedia of Traditional Chinese Medicine (ETCM)), shared gene targets were extracted using Metascape (metascape.org/gp/index.html) and the Kyoto Encyclopedia of Genes and Genomes (KEGG; https://www.genome.jp/kegg; 15 January 2022). The database was analyzed for gene ontology (GO) enrichment and pathway enrichment of these potential targets.

### 2.3. Animal Experiments

Specific-pathogen-free, 5-week-old male C57BL/6/J mice were purchased from Vital River Laboratories (Beijing, China). All animal protocols used in this study were approved by the Animal Experimentation Ethics Committee of Jiangnan University, Wuxi, China (JN. No2016102520170411(76)). All experiments were performed at the Experimental Animal Center of Jiangnan University. After 1 week of adaptation, the mice were randomly divided into a normal-diet control group (10% fat diet; Beijing HFK Bioscience Co., Ltd., Beijing, China) and two high-fat diet groups (HF and MFE groups; 60% fat diet; Beijing HFK Bioscience Co., Ltd.), which were both fed a high-fat diet for 13 weeks to establish a NAFLD model. An MFE dose of 125 mg/kg was chosen as the optimal dose for these experiments as this dose was shown to have minimal side effects. A gavage solution was prepared by dissolving MFE in 0.5% sodium carboxymethylcellulose (CMC; Sinopharm Chemical Reagent Co., Ltd., Shanghai, China). One of the high-fat diet groups (MFE group) was administered MFE by oral gavage daily for 4 weeks. The other two groups of mice were treated with an equal volume of 0.5% CMC. The food intake of the mice was observed, and the body weight and fasting blood glucose concentrations were measured weekly. After the mice were euthanized, their livers were weighed, and blood, feces, and tissue samples were collected and stored at −80 °C until further analysis.

### 2.4. Serum Biochemical Markers in Mice

Blood samples were collected from mice in the dark and left to coagulate at room temperature (25 °C) for 1 h. After centrifuging the blood samples at 3500 rpm for 10 min, serum samples were harvested into an Eppendorf tube and stored at −80 °C in a deep freezer until further analysis. Serum triglyceride (TG), total cholesterol (TC), low-density lipoprotein cholesterol (LDL-c), high-density lipoprotein cholesterol, alanine aminotransferase (ALT), and aspartate aminotransferase (AST) concentrations were measured using an automatic biochemical analyzer (Mindrayn BS-480; Mindray, Shenzhen, China).

### 2.5. Hematoxylin & Eosin and Oil Red O Staining

Fresh epididymal fat and liver tissue samples were collected, fixed in 4% paraformaldehyde in phosphate-buffered saline for 2–4 h at 4 °C, and then embedded in paraffin. Tissue sections were stained with hematoxylin and eosin (H&E). For Oil Red O staining, fresh liver tissue was snap-frozen in liquid nitrogen. Sections were cut at a low temperature and stained with Oil Red O and H&E. Stained tissue sections were imaged using a digital slide scanner (Panoramic MIDI II; 3DHISTECH Ltd., Budapest, Hungary) and analyzed using Image-Pro Plus software (Media Cybernetics, Rockville, MD, USA). The rest of the tissue was stored at −80 °C.

### 2.6. Quantitative Reverse Transcription Polymerase Chain Reaction Analysis

Total RNA was isolated from ileal and liver tissues using TRIzol (Invitrogen, Carlsbad, CA, USA). RNA samples (1 µg) were reverse transcribed using HiScript III-RT SuperMix (+gDNA wiper; Vazyme, Nanjing, China) to generate complementary DNA. Gene expression was qualitatively analyzed using polymerase chain reaction (PCR) and agarose gel electrophoresis. Quantitative (q) PCR was performed using a CFX Connect real-time system (Bio-Rad, Hercules, CA, USA), and the results were analyzed using the 2-ΔΔCt method.

### 2.7. Gut Microbiota Analysis by 16S rRNA Gene Sequencing

Mouse feces were collected in the 13th experimental week and DNA was extracted using the FastDNA® Spin Kit for Feces (MP Biomedicals, Santa Ana, CA, USA), according to the manufacturer’s instructions. DNA samples were then purified using the TIANgel Mini Purification Kit (TIANGEN, Beijing, China) and quantified using the Qubit dsDNA HS Assay Kit (Life Technologies Corporation, Carlsbad, CA, USA). Samples were sequenced using previously described methods [27]. The sequencing data were analyzed using Microbiomeanalyst (https://www.microbiomeanalyst.ca/; 15 January 2022) and Galaxy (http://huttenhower.sph.harvard.edu/galaxy/; 15 January 2022) software tools.

### 2.8. Non-Targeted Metabolomics Analysis

Liquid chromatography (U3000; Thermo Scientific, Waltham, MA, USA)–mass spectrometry (Q Elective, Thermo Scientific) was performed using a C18 column, 0.1% *v*/*v* formic acid in water for mobile phase A, and acetonitrile under positive ion conditions for mobile phase B. The flow rate was 0.3 mL/min and the sample volume was 5 μL. The following gradient elution program was used: 0–3 min: 100% A, 3–13 min: 100–20% A, 13–16.5 min: 20% A, 16.5–20 min: 100% A (for column elution). The mass spectrometry detection conditions were as follows: resolution, 35,000; scan range, 70–1050 *m*/*z*; electrospray voltage, 3.5 kV; electron ionization energy, 20, 40, and 60 eV. A quality control sample was prepared by pooling the aliquots of the study samples and was used to monitor instrument stability. Substance extraction, quantification, and relative abundance analyses of the raw data output of the liquid chromatography–mass spectrometry experiments were performed using Compound Discoverer 3.1 (Thermo Scientific, Waltham, MA, USA).

The mass spectra information of the extracted metabolites was used to search the databases, Massbank (https://massbank.eu), PubChem (http://ncbi.nlm.nih.gov/), Chemspider (www.chemspider.com), and mzCloud (https://www.mzcloud.org), to obtain structural information for the substances. Principal component analysis (PCA) and Spearman’s correlation analysis were performed using MetaboAnalyst 5.0 (https://www.metaboanalyst.ca). If the results of the Spearman’s correlation analysis showed *p* < 0.05, the variables were considered to be correlated. Comparative analysis between groups and pathway enrichment and prediction were performed using OmicsBean (www.omicsbean.cn). Between-group comparisons were performed using a two-tailed Student’s *t* test (two groups) or one-way or two-way analysis of variance ANOVA (multiple groups, adjustments for pairwise comparisons).

### 2.9. Statistical Analysis

All data are presented as means ± standard deviations. Data from the animal experiments were analyzed by one-way ANOVA. Multiple comparisons between groups were performed using Dunnett’s test. For all of the statistical tests, the 95% confidence interval or *p* < 0.05 was used as the threshold for significance.

## 3. Results

### 3.1. MFE Principal Component Analysis and Identification

We analyzed the main components of MFE by GC–MS, and the results showed that lignans such as licarin A, licarin B, and otobain accounted for the main part (17.77%), polysaccharides (2.69%) such as melezitose and sucrose, and fatty acids such as tetradecanoic acid (8.57%) were present in the extract. In addition, flavonoids such as methyl eugenol (2.84%) and terpenoids such as cedrol (1.72%) were also detected (Table 1).

### 3.2. Alleviation Effect of MFE on High-Fat-Diet-Induced NAFLD in Mice

The effect of the MFE was determined in mice with high-fat-diet-induced NAFLD. We examined the daily food intakes, body weights, and liver weights of the mice in the three groups (Table 2). Food intake by the mice did not change significantly, but the body weights and liver weights of the mice were significantly reduced after MFE treatment. Serum biochemical index results showed that MFE effectively decreased blood lipid concentrations (LDL-c, TC, and TG); improved liver function, as indicated by decreased AST and ALT concentrations; and decreased fasting blood glucose concentrations (Table 3). As shown in Figure 1A,B, both H&E- and Oil-Red-O-stained sections showed that hepatic lipid accumulation was significantly reduced in the MFE group compared with the HF group. Finally, we detected levels of inflammatory factors such as tumor necrosis factor (TNF)-α, interleukin (IL)-6, and IL-1β in the livers of the mice (Figure 1C–E). The level of liver inflammation was significantly reduced in the MFE group compared with the HF group. Thus, MFE significantly regulates lipid metabolism and inflammation in mice fed a high-fat diet.

### 3.3. Modulation of the Gut Microbiota by MFE in Mice Fed a High-Fat Diet

To further investigate the effect of MFE on the gut microbiota, we analyzed changes in the microbiota structure in mouse feces after MFE treatment by using 16S rRNA analysis. MFE effectively regulated the abundance of Bacteroidetes and Firmicutes, with the proportion of Bacteroidetes increasing to approximately the proportion seen in the control group (Figure 2A). Partial least-squares discriminant analysis (PLS-DA) was then used to analyze the differences in the microbiota between the three groups (Figure 2B). We found that microbial diversity was significantly different in the HF group compared with the MFE and control groups along the t2 axis. However, there was an intersection between the MFE group and the control group. Compared with the HF group, the Shannon index value from the alpha diversity analysis was significantly higher in the MFE group (Figure 2C). We found that the probiotic species typically used to treat NAFLD, such as members of the genera Lactobacillus, Akkermansia, and Bacteroides, were present in the core microbiome composition (Figure 2D). At the genus level, the abundance of probiotic bacteria, such as Akkermansia, Blautia, Bifidobacterium, and Adlercreutzia, was significantly increased in the MFE group compared to the HF group (Figure 2E–H).

In the phylogenetic tree, the distance between samples in the HF group and the MFE group was greater than that between the other two groups, and the two groups were located on different branches (Figure 3A). In addition to the genera mentioned above (Figure 2E–H), the abundance of the genera Lactococcus and SMB53 were also significantly increased in the MFE group, as shown in the differential enrichment heat map (Figure 3B). Linear discriminant analysis effect size analysis showed that genera from the families Clostridiaceae and Peptostreptococcaceae and the genus Bacteroides were predominant in the MFE group, whereas the class Clostridiales and the genera Allobaculum and Odoribacter were predominant in the HF group (Figure 3C). A circular cladogram shows the differentially abundant taxa in the three groups (Figure 3D). We also found that Akkermansia, Blautia, and Bifidobacterium were closely related to other genera and the MFE group had the largest proportion of the three genera in the interaction network diagram (Figure 3E). These results showed that MFE improved the structure of intestinal flora and increased the abundance of probiotic species. 

### 3.4. Effects of MFE on Intestinal Metabolites in Mice Fed a High-Fat Diet

We further analyzed the metabolites in the mouse feces using off-target metabolomics. Through PLSDA analysis and the construction of a phylogenetic tree, we found that the metabolite profiles did not significantly differ between the HF and MFE groups, and there were overlapping components between the two groups. However, the metabolite profiles of the HF and MFE groups were significantly different from the metabolite profile of the control group and these groups showed a certain degree of separation (Figure 4A,C). This may be due to the long-term effects of a high-fat diet and the relatively short duration of MFE treatment (4 weeks).

Indole (C00463), indole-3-acetic acid (C00954), and tryptophan (C00525), which are related to tryptophan metabolism, were significantly enriched in the MFE group compared with the HF group. (Figure 4B). Pathway enrichment analysis showed that the metabolic pathways in the MFE group were mainly enriched in tryptophan metabolism; phenylalanine, tyrosine, and tryptophan biosynthesis; and thiamine metabolism, as indicated in the bubble chart (Figure 4D). Here, we mainly focus on the enriched metabolites associated with tryptophan synthesis and metabolism (Figure 4E). These findings indicate that MFE regulated metabolites in the mouse gut, mainly through the tryptophan metabolic pathway.

### 3.5. Omics Analysis of the Intestinal Flora Metabolites in NAFLD Mice after MFE Treatment

Spearman’s correlation analysis was used to investigate the correlations between the gut microbiota composition and metabolite levels. PLSDA analysis of the flora (abscissa) and the metabolites (ordinate) was used to determine the sample distribution, which was consistent with the aforementioned microbiome and metabolomics results (Figure 5A). We then visualized the correlations between metabolite levels and microbiota composition using a heat map (Figure 5B). The abundance of Akkermansia, Blautia, and Bifidobacterium was significantly positively correlated with tryptophan and indole-3-acetic acid levels, and negatively correlated with 5-hydroxyindole-3-acetic acid levels. The negative correlation was significant between Bifidobacterium and 5-hydroxyindole-3-acetic acid. In contrast, the abundance of Bacteroidetes and S24-7 was significantly negatively correlated with tryptophan and indole-3-acetic acid levels, respectively. As shown in Figure 5C, we used the network interaction map to more intuitively analyze the correlation between the flora composition and metabolite levels. It was clear that tryptophan and indole-3-acetic acid interacted with multiple genera, and therefore, this metabolic pathway is closely related to the composition of the gut flora.

### 3.6. MFE Alleviates Hepatic Fat Accumulation and the Inflammatory Response by Regulating the Aryl Hydrocarbon Receptor–Fatty Acid Synthase/Sterol Regulatory Element-Binding Protein 1c and Nuclear Factor Kappa B Signaling Pathways in Mice

We extracted two data sets (GSE151158 and GSE57425) related to NAFLD from the GEO database to analyze differential gene expression between healthy individuals/mice and patients/mice with NAFLD. Data from both datasets were normalized (Figure 6A, GSE151158; Figure 6B, GSE57425). Volcano plots were used to identify DEGs in GSE57425 (Figure 6C; red: upregulated; blue: downregulated). PCA was used to determine differences in gene expression levels between the samples in each group (for example, GSE57425 in Figure 6D). Significant differences in the gene expression profiles between groups are shown as a heat map (Figure 6E). Cytochrome P450 family 7 subfamily B member 1, IL-1 receptor type 1, and TNF-α-induced protein 3 were expressed at significantly higher levels in the NAFLD group compared with the normal group. Cytochrome P450 family 17 subfamily A member 1 and cathelicidin antimicrobial peptide were significantly downregulated in the NAFLD group compared with the normal group. The expression levels of these genes are closely related to liver inflammation.

We mined *M. fragrans* Houtt-related gene targets in the Traditional Chinese Medicine System Pharmacology Database and Analysis Platform (TCMSP) and the Encyclopedia of Traditional Chinese Medicine (ETCM) databases using the following criteria: OB ≥ 30%, BBB permeability ≥ −0.3, and DL ≥ 0.18, and we matched the target of each component. The gene targets identified were matched with data from the GEO database, and 53 gene targets were obtained (Figure 7A). GO and pathway analyses were performed using.

Metascape and KEGG databases, respectively. GO analyses revealed the following significant terms: immune response (GO: 0006955), innate immune response (GO: 0045087), inflammatory response (GO: 0006954), and adaptive immune response (GO: 0002250) in the biological process category, and plasma membrane (GO:0005886) in the cellular component category. In Figure 7C, the Sankey diagram on the left shows the relationship between the five most enriched pathways and the related gene targets. The bubble chart on the right shows the enrichment of each pathway. The nuclear factor kappa B (NF-κB) signaling pathway was the most enriched pathway, and of the members of this pathway, the targets TNF, IL1B, and inhibitor of NF-κB kinase subunit beta (IKBKB) were significantly enriched. Multiple additional pathways were also enriched. Finally, we verified the targets in the NF-κB pathway related to liver inflammation and the aryl hydrocarbon receptor (AhR)–fatty acid synthase (FAS) signaling pathway related to intestinal tryptophan metabolism (Figure 7D–K).

In this study, IKBKB and AhR were significantly upregulated in the MFE group compared with the HF group. NF-κB subunit 1 (NFKB1), TNF, IL-6, IL-1b, FAS, and sterol regulatory element-binding transcription factor 1 (SREBP-1c) were significantly downregulated by MFE.

## 4. Discussion

MEFs (Ingredients in *M. fragrans* extract) are considered to have potential as prebiotics or regulators of gut health (Table 1). Myristic acid, a natural edible spice, is thought to be effective in alleviating inflammation in diabetic rats [28]. Methyl eugenol has also been shown to have analgesic and anti-inflammatory effects [29]. Melezitose and sucrose can be used by the gut flora as carbon sources. It is worth noting that the lignans in the licarin A, licarin B, and otobain families have potential efficacy in regulating the gut microbiota and in the treatment of NAFLD. In this study, we treated mice fed a high-fat diet with MFE and found reduced hepatic lipid accumulation and improved liver function. Although previous studies reported that nutmeg extracts, such as myrislignan, have better anti-inflammatory effects in the liver [25], these studies did not consider the metabolism and degradation of these extracts in the digestive tract after oral administration, or their processing and utilization by intestinal flora. Many natural products that protect the liver or cause weight loss, such as berberine, have low oral bioavailability, and must be processed by the large intestine and the intestinal flora [30]. Therefore, studies that ignore the role of the gut microbiota and their metabolites when studying the direct targets of drug components are insufficient. Here, for the first time, we investigated the effect of MFE on the gut microbiota and its metabolites in mice fed a high-fat diet.

In studies of the regulation of the gut microbiota by MFE, a variety of probiotics were found to be upregulated by MFE, and MFE was shown to play the role of a prebiotic. Akkermansia species are considered to be second-generation probiotics, with important contributions to lipid metabolism [31,32]. As shown in Figure 2E, we found that the abundance of Akkermansia was approximately the same in the MFE group and the control group. This indicates that MFE may help maintain normal levels of Akkermansia in mice fed a high-fat diet. Blautia may be a new probiotic [33]. The abundance of species in the genus Blautia are significantly reduced in obese children [34]. As shown in Figure 2F, the abundance of Blautia was higher in the MFE group than the control group. The same was true for Bifidobacterium and Adlercreutzia. Bifidobacterium species are widely used to promote health and treat disease [35,36]. In recent years, Adlercreutzia has been found to be a probiotic associated with disorders of lipid metabolism. It has been reported that the burning of fat is accelerated at low temperatures, and the abundance of Adlercreutzia increases under these conditions [37]. A correlation analysis of microflora network interactions showed that probiotic species, such as those in the genera Blautia and Bifidobacterium, had more neighbors, and the MFE group exhibited the highest abundance of these bacteria of all three experimental groups. This also shows that these genera play important roles in the intestine.

Metabolite enrichment analysis of the gut microbiota showed that tryptophan metabolism was the most important metabolic pathway. Tryptophan follows three main metabolic pathways: the kynurenine [38], serotonin [39], and microbial metabolic pathways [40]. Although the first two pathways affect appetite through the brain–gut axis, resulting in weight loss, no changes in appetite were observed after MFE treatment in this study. Thus, we focused our attention on the microbial pathway. AhR is a xenobiotic sensor that is known to be an important regulator of metabolic and immune processes [41]. In addition to well-known man-made pollutants (e.g., 2,3,7,8-tetrachlorodibenzo-p-dioxin) [42], a battery of natural AhR ligands have been discovered.

The metabolites of tryptophan, indole, indole triacetate, and tryptamine are ligands for AhR and are responsible for activating AhR and regulating downstream targets [43]. Moreover, enzymes that metabolize tryptophan, such as bacterial tryptophanase, are produced by intestinal bacteria [44]. The abundance of the genus Bacteroides was found to be negatively related to tryptophan levels. This may be because Bacteroides are rich in tryptophanase, which is involved in tryptophan metabolism. The abundance of Akkermansia, Blautia, and Bifidobacterium was positively correlated with tryptophan levels, indicating that there were more probiotics and more tryptophan in the MFE group to activate AhR. A previous study suggested that tryptophan metabolites of the gut microbiota suppress liver inflammation [7], and the 3-indoleacetic acid produced by the intestinal flora helps mice lose weight [45].

Through GEO database mining, we obtained the gene expression profiles of individuals/animals on a high-fat diet or with NAFLD, which provided the direction for subsequent differential gene expression analysis. Through an analysis of the ETCM database, we identified disease targets corresponding to the MFE components and matched these data with data from the GEO database to obtain potential targets of the MFE in the treatment of NAFLD. Based on the results of enrichment analysis, we focused on the NF-κB signaling pathway. Finally, after qPCR verification, we came to the conclusion that MFE inhibited the nuclear factor kappa B (NF-κB) pathway and downregulation of the aryl hydrocarbon receptor (AhR)–mediated fatty acid synthase (FAS)–sterol regulatory element-binding protein 1c pathway.

In conclusion, MFE alleviated NAFLD by modulating the intestinal flora, especially the abundance of probiotic species; by improving the intestinal environment; by regulating intestinal tryptophan metabolism; by activating AhR in the liver; and by regulating the expression of the downstream targets FAS and SREBP-1c. MFE also regulated hepatic inflammation by regulating the NF-κB signaling pathway, and its upstream factor, IKBKB, was activated to inhibit the expression of NF-κB, thereby downregulating the expression of the downstream cytokines TNF, IL-6, and IL-1b to reduce the level of liver inflammation. 

## Figures and Tables

**Figure 1 nutrients-14-01699-f001:**
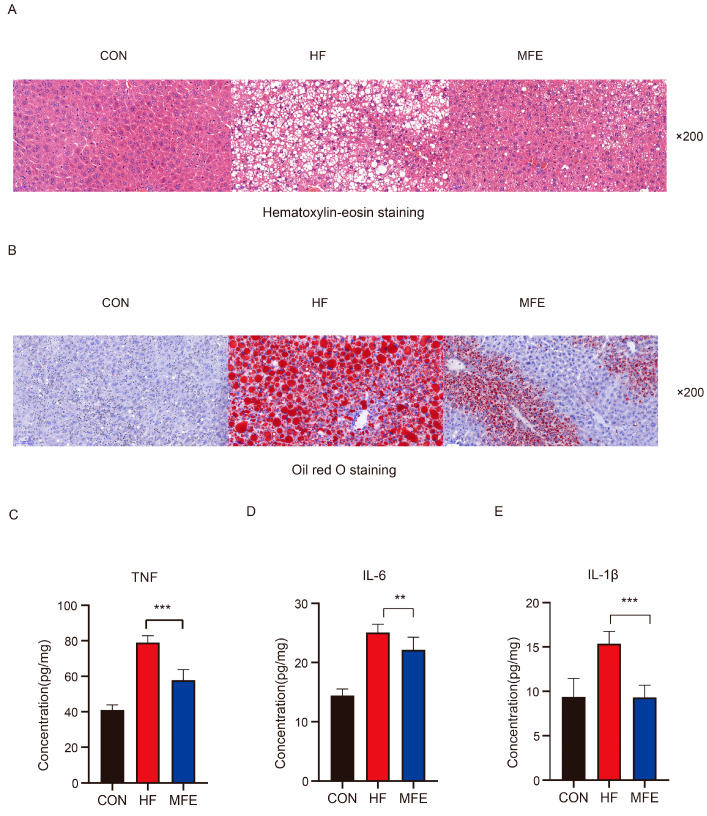
Effects of MFE on hepatic lipid accumulation and inflammatory factor levels in high-fat-diet mice. (**A**) Hematoxylin-eosin staining (H&E) of mice liver section, 200× magnification. (**B**) Oil Red O staining of mice liver section, 200× magnification. (**C**) Concentration levels of TNF-α in the liver. (**D**) Concentration levels of Il-6 in the liver. (**E**) Concentration levels of IL-1β in the liver. Data are shown as mean ± SD. (For each group, *n* = 10, ** *p* < 0.01; *** *p* < 0.001).

**Figure 2 nutrients-14-01699-f002:**
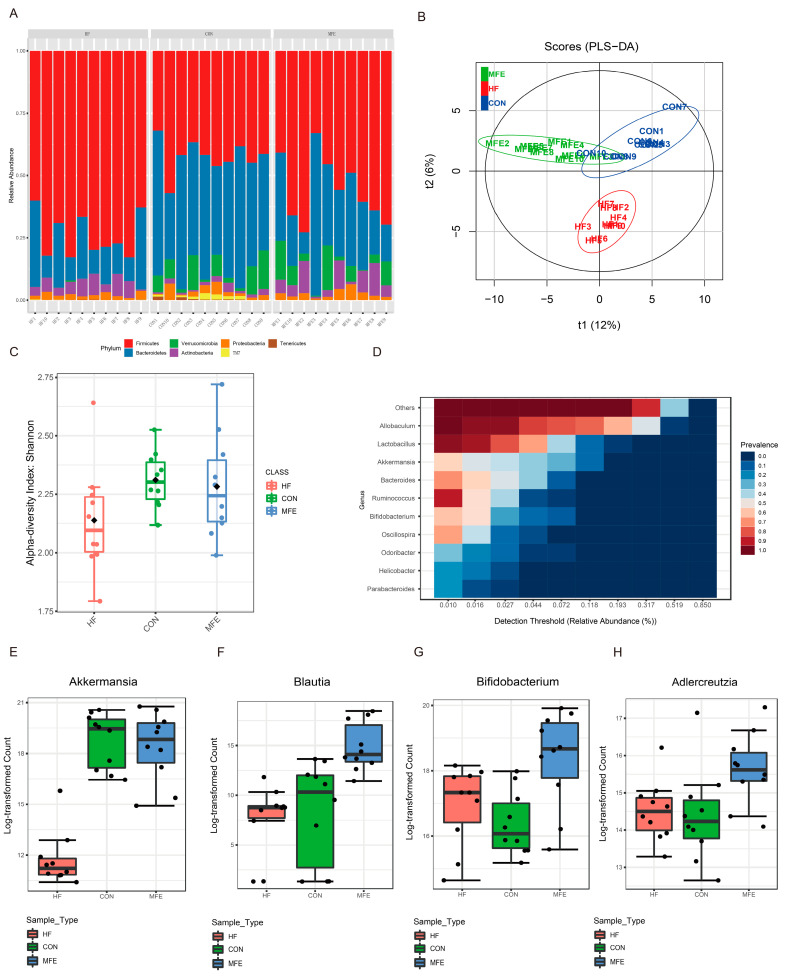
Effects of MFE on the abundance and diversity of gut microbiota. (**A**) Bacteria composition at the phylum level in each sample (top7). (**B**) Partial least-squares discriminant analysis (PLS-DA) of beta diversity in three groups (CON, HF, and MFE). (**C**) The Shannon diversity index of alpha diversity in three groups (CON, HF, and MFE). (**D**) Correlation analysis of the core gut bacteria at the genus level. (**E**–**H**) The bacteria ((**E**) Akkermansia, (**F**) Blautia, (**G**) Bifidobacterium, and (**H**) Adlercreutzia) were significantly upregulated by MFE at the genus level in high-fat-diet mice. Data are shown as mean ± SD.

**Figure 3 nutrients-14-01699-f003:**
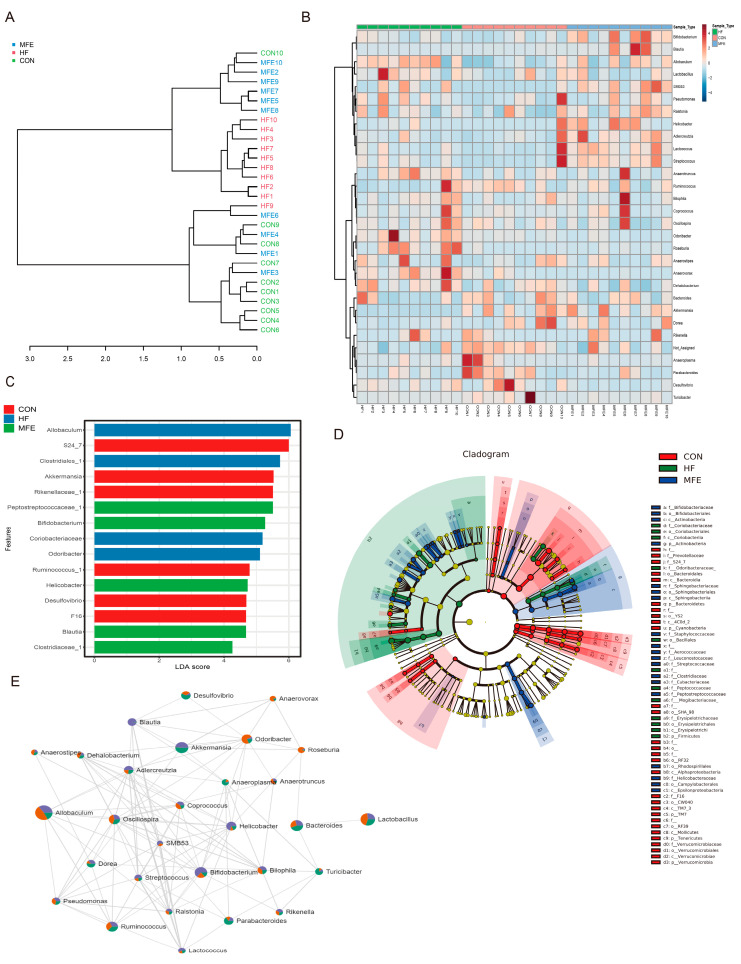
Correlation and difference analysis of intestinal flora via MFE in mice fed a high-fat diet. (**A**) Phylogenetic tree of correlation between samples at genus level of three groups (CON, HF, and MFE). (**B**) Heat map of the correlation between samples at the genus level (Top30) of three groups. (**C**) Linear Discriminant Analysis (LDA) effect size (LEfSe) analysis at OTU level in three groups. (**D**) A circular cladogram was generated to show the differentially abundant taxa. (**E**) Genus-level gut microbe network diagram of three groups (green: CON, orange: HF, and blue: MFE) to show correlations between genera and proportions among groups. Data are shown as mean ± SD. (For each group, *n* = 10).

**Figure 4 nutrients-14-01699-f004:**
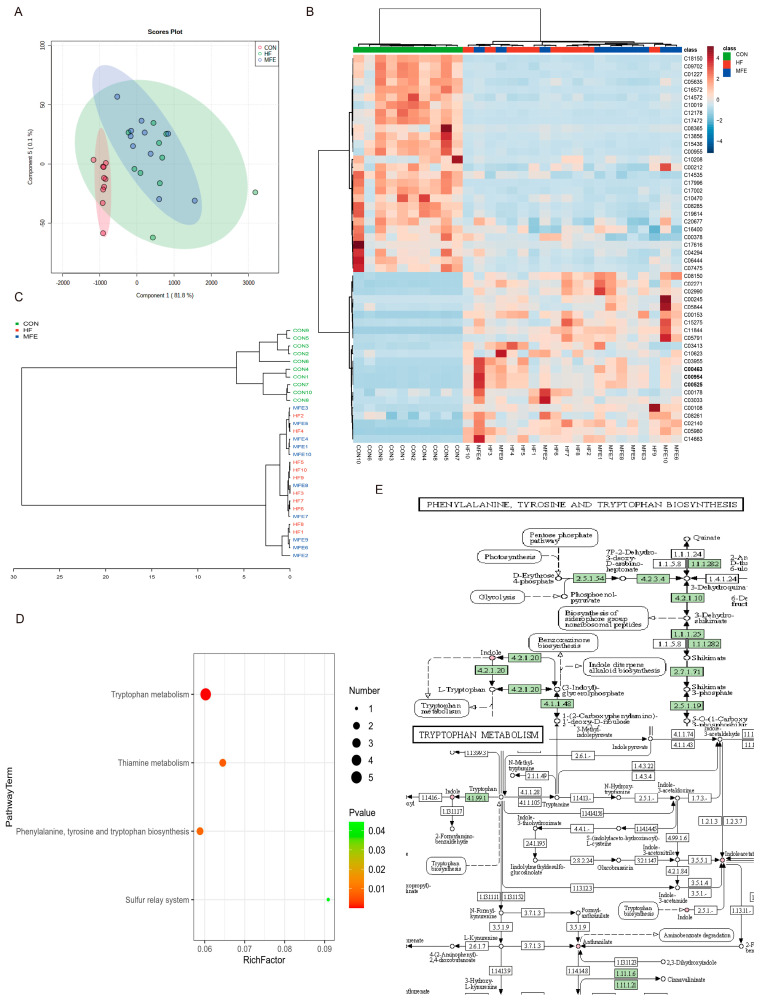
Effects of MFE on gut metabolites. (**A**) Partial least-squares discriminant analysis (PLS-DA) of metabolites of three groups (CON, HF, and MFE). (**B**) Heat map for metabolite enrichment change analysis (Top50) of three groups. (**C**) Phylogenetic tree of correlation between samples of metabolites in three groups (CON, HF, and MFE). (**D**) The pathway enrichment analysis of metabolites. (**E**) KEGG map shows metabolites in enriched pathways. Data are shown as mean ± SD. (For each group, *n* = 10).

**Figure 5 nutrients-14-01699-f005:**
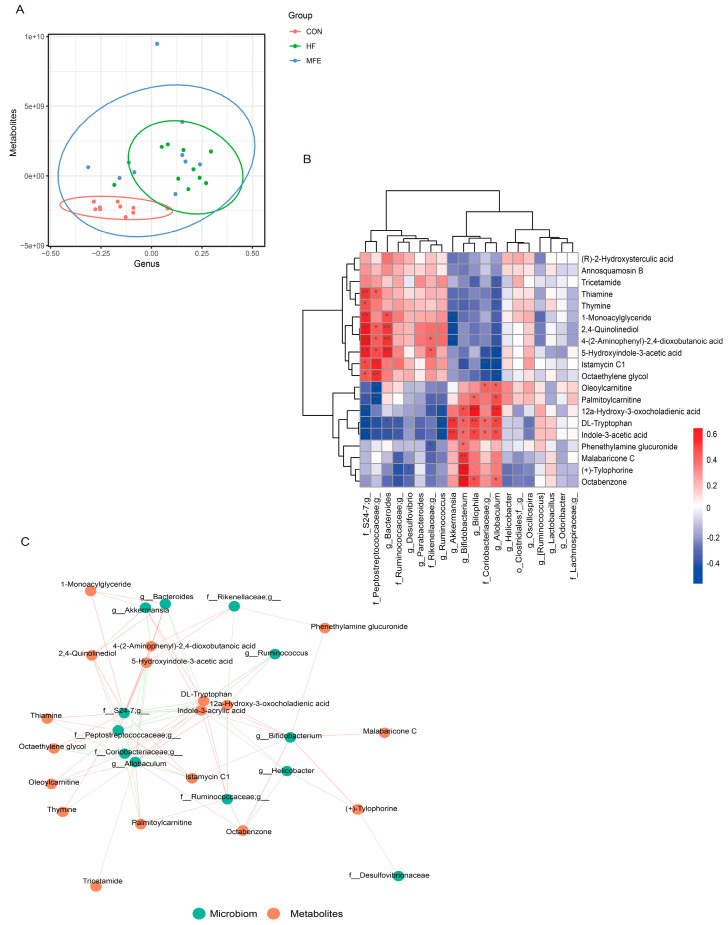
Omics analysis of the intestinal flora metabolites in NAFLD mice with MFE treatment. (**A**) PLS-DA (Partial Least Squares Discriminant) Analysis of intestinal microbes at genus level and metabolites. (Variable Importance for the Projection, VIP > 1) (**B**) Heat map of correlation analysis between intestinal microbes at genus level and metabolites in MFE group (red represents positive correlation and blue represents negative correlation). (**C**) Correlation network diagram between microbiota and metabolites in MFE group. Data are shown as mean ± SD. (For each group, *n* = 10, * *p* < 0.05; ** *p* < 0.01; *** *p* < 0.001).

**Figure 6 nutrients-14-01699-f006:**
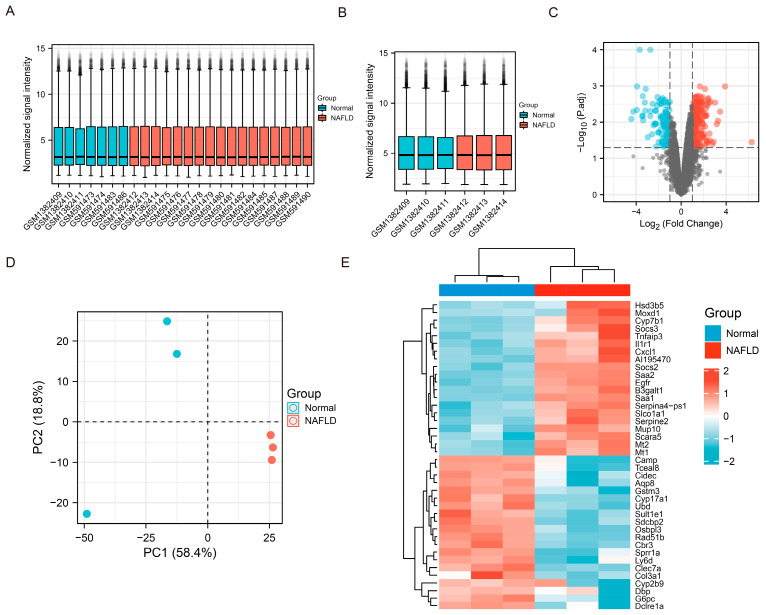
Mining and Analysis of GEO (Gene expression omnibus) database related to NAFLD. (**A**,**B**) The normalization of data sets (GSE151158 GSE57425). (**C**) Volcano plot of differential gene analysis (|logFC| > 1; *p*-value < 0.05). (**D**) Principal component analysis (PCA) of each group of samples in the GSE57425 data set. (**E**) Heat map of top 20 gene expressions (red: high expression; blue: low expression) in the expression profile. Data are shown as mean ± SD.

**Figure 7 nutrients-14-01699-f007:**
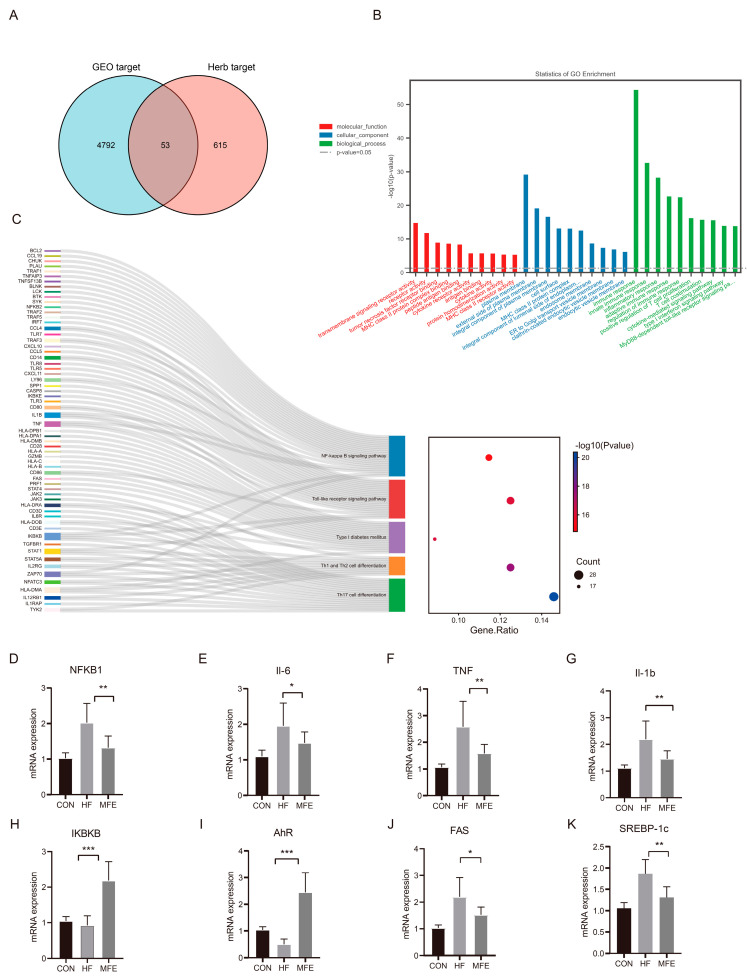
Effects of MFE on NF-κB pathway related to liver inflammation and the AhR–FAS signaling pathway related to the intestinal tryptophan metabolism pathway. (**A**) The intersection of the targets of the GEO database and the targets of the Chinese herbal medicine database. (**B**) Gene ontology (GO) analysis of 53 targets. (**C**) KEGG pathway analysis (**D**–**K**) The targets of NF-κB pathway signaling (NKFB1, Il-6, TNF, Il-1b, and IKBKB) and The targets of AhR–FAS signaling (AhR, FAS, and SREBP-1c). (For each group, *n* = 10, * *p* < 0.05; ** *p* < 0.01; *** *p* < 0.001).

**Table 1 nutrients-14-01699-t001:** The main ingredients of *Myristica fragrans* extracts (MFE).

Ingredient	Content (%)
Tetradecanoic acid	8.57
Licarin A	9.94
Licarin B	6.46
Methyleugenol	2.84
Isoelemicin	2.55
Cedrol	1.72
Melezitose	2.14
Sucrose	1.32
Otobain	1.37

**Table 2 nutrients-14-01699-t002:** The effect of MFE on food intake and body and liver weight in high-fat-diet mice.

	CON	HF	MFE
Food intake (g/day)	2.92 ± 0.27 ^a^	2.74 ± 0.21 ^a^	2.94 ± 0.08 ^a^
Final body weight (g)	31.61 ± 1.09 ^c^	39.54 ± 1.39 ^a^	34.27 ± 1.25 ^b^
Body weight gain (g)	12.71 ± 1.41 ^c^	21.25 ± 1.27 ^a^	16.01 ± 1.29 ^b^
Liver weight (g)	1.29 ± 0.19 ^c^	1.94 ± 0.21 ^a^	1.56 ± 0.17 ^b^
Liver index (%)	4.09 ± 0.57 ^ab^	4.96 ± 0.59 ^a^	4.53 ± 0.42 ^a^

Data are shown as mean ± SD. The values with different letters (a–c) are significantly different (*p* < 0.05) between the indicated groups.

**Table 3 nutrients-14-01699-t003:** The effect of MFE on serum biochemical indicators in high-fat-diet mice.

	CON	HF	MFE
TC (U/L)	2.15 ± 0.39 ^b^	3.59 ± 0.51 ^a^	2.44 ± 0.57 ^b^
TG (U/L)	0.45 ± 0.07 ^b^	0.72 ± 0.09 ^a^	0.51 ± 0.08 ^b^
HDL-c (mmol/L)	1.91 ± 0.23 ^a^	1.99 ± 0.13 ^a^	2.10 ± 0.31 ^a^
LDL-c (mmol/L)	0.46 ± 0.18 ^b^	0.93 ± 0.21 ^a^	0.47 ± 0.11 ^b^
AST (mmol/L)	65.76 ± 10.52 ^c^	130.98 ± 16.75 ^a^	90.98 ± 13.99 ^b^
ALT (mmol/L)	54.15 ± 16.9 ^b^	198.33 ± 30.78 ^a^	75.35 ± 23.17 ^b^
Glu (mmol/L)	4.89 ± 0.5 ^c^	6.97 ± 0.89 ^a^	5.85 ± 0.83 ^b^

Data are shown as mean ± SD. The values with different letters (a–c) are significantly different (*p* < 0.05) between the indicated groups.

## Data Availability

Data is contained within the article or Appendix A. The data presented in this study are available in [insert article or Appendix A here].

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
