# Peer review of "Myristica fragrans Extract Regulates Gut Microbes and Metabolites to Attenuate Hepatic Inflammation and Lipid Metabolism Disorders via the AhR–FAS and NF-κB Signaling Pathways in Mice with Non-Alcoholic Fatty Liver Disease"

_nutrients, 2022, doi:10.3390/nu14091699_

Round 1
Reviewer 1 Report
The manuscript presented by Zhao and collaborators brings important and new scientific results about the effect of nutmeg on health, which were explained by microbiome changes according their results. The manuscript is well written and the methodology is clear, as well the results. However, I pointed out some specific considerations that shoud be addressed by the authors.
Introduction section: The relevance and the differential of the study should be more consistent. And the main objetive should be presented, which is not clear.
Material and Methods section: The animal management should be detailed, the same idea for the metabolites data analysis. The bioinformatic pipeline applied, parameters of normalization used, for example. How many animals were used in each treatment?
Results section: The label of the Figures are small and when I tried to zoom in the quality of the image was unreadable.
Table 1 is on top of the text.
I did not understand the difference of the number of samples considered in the Figures 6A, 6B and 6D. Please explain it.
In general, the authors should detailed their data analysis and present the raw results of microbiome as supplemental material.
My main concerned and disagreement on the manuscript was the use of the expression gene data obtained from the public database to explain the effect of the nutmeg effect on gene expression level. Because the nutmeg probably affects the gene expression level of the different tissues not only the microbiome. The authors should validate it on their on samples.
Reviewer 2 Report
In this study, the authors have examined the role of Myristica fragrans Extract on NAFLD and discovered a regulatory role of MFE on hepatic inflammation, gut microbes, and metabolites. The authors have applied various types of bioinformatics tool for the analysis. Although some findings are of novelty, results do not seem to be connected logically to prove the authors' hypothesis.
1) What is the connection between the findings in gut microbiota and amelioration of fatty liver? There is no actual evidence on whether improved microbiota is linked to improved liver.
2) Line 78: 'Antibiobics, obesity, and lipid-lowering drugs have adverse effects on gut microbiota' - this sentence does not seem to be related to the sentence before and after.
3) Line 112: I believe the name is GEO2R not GEOR2
4) Ling 115: logFC>1 and p <0.05 seems to be a very generous criteria. It is also recommended to use FDR value for significance.
5) It is not very clear how 'Drug data' from the herbal database is can be useful here, since it is too obvious that NF-kB is involved in NAFLD.
6) To confirm their animal study design as NAFLD model, score measure should be done. The biomarkers shown here only proves fatty liver. AST and ALT measurement is not enough.
7) How was the liver index(%) calculated?
8) It is not clear which data is from GSE57425 and which data is from GSE151158 in the figures.
Round 2
Reviewer 1 Report
All my considerations were answered. I don´t have additional comments or questions.
Reviewer 2 Report
Although some comments were not fully answered, the revised manuscript is sufficiently improved for publication.